# Enhancing materials property prediction by leveraging computational and experimental data using deep transfer learning

Dipendra Jha[1], Kamal Choudhary[2], Francesca Tavazza[2], Wei-keng Liao[1], Alok Choudhary[1], Carelyn Campbell[2] & Ankit Agrawal[1]*

The current predictive modeling techniques applied to Density Functional Theory (DFT) computations have helped accelerate the process of materials discovery by providing significantly faster methods to scan materials candidates, thereby reducing the search space for future DFT computations and experiments. However, in addition to prediction error against DFT-computed properties, such predictive models also inherit the DFT-computation discrepancies against experimentally measured properties. To address this challenge, we demonstrate that using deep transfer learning, existing large DFT-computational data sets (such as the Open Quantum Materials Database (OQMD)) can be leveraged together with other smaller DFT-computed data sets as well as available experimental observations to build robust prediction models. We build a highly accurate model for predicting formation energy of materials from their compositions; using an experimental data set of $1,963$ observations, the proposed approach yields a mean absolute error (MAE) of $0.06$ eV/atom, which is significantly better than existing machine learning (ML) prediction modeling based on DFT computations and is comparable to the MAE of DFT-computation itself.

[1] Department of Electrical and Computer Engineering, Northwestern University, Evanston, IL 60208, USA. [2] Thermodynamics and Kinetics Group, National Institute of Standards and Technology, Gaithersburg, MD 20899, USA. *email: ankitag@eecs.northwestern.edu

Experimental observations have been the primary means to learn and understand various chemical and physical properties of materials[1–6]. Nevertheless, since experiments are expensive and time-consuming, materials scientists have been relying on computational methods such as Density Functional Theory (DFT)[7] to compute materials properties and model processes at the atomic level to help guide experiments[8]. DFT has enabled the creation of high-throughput atomistic calculation frameworks for accurately computing (predicting) the electronic-scale properties of a crystalline solid using first principles, which can be expensive to measure experimentally. Over the years, such DFT computations have led to a number of large data sets like the Open Quantum Materials Database (OQMD)[9,10], the Automatic Flow of Materials Discovery Library (AFLOWLIB)[11], the Materials Project[12–14], Joint Automated Repository for Various Integrated Simulations (JARVIS)[15–18], and the Novel Materials Discovery (NoMaD)[19]. They contain DFT-computed properties of $\sim 10^4$–$10^6$ materials, which are either experimentally-observed[20] or hypothetical materials. The availability of such large DFT-computed data sets has spurred the interest of materials scientists to apply advanced data-driven machine learning (ML) techniques to accelerate the discovery/design of new materials with select engineering properties[21–45]. Such predictive models enable reducing the size of the search space for material candidates and help in prioritizing which DFT simulations and, possibly, experiments, to perform. Training data sizes can have significant impact on the quality of prediction performance in ML and particularly in deep learning[46]. This has also been proven specifically for the case of the prediction of material properties[33,42]. As experimental data are limited in materials science,

ML models are mostly trained using DFT-computational data sets[24,32,33,42,47–49].

Some recent works compare the DFT-computed formation energies with experimental observations[10,50,51]. For instance, Kirklin et al. compared the DFT-computed formation energy with experimental measurements of 1670 materials and found the mean absolute error (MAE) to vary from 0.096 to 0.136 eV/atom for OQMD[10]. Jain et al.[51] reports the MAE of the Materials Project as 0.172 eV/atom, whereas in Kirklin et al.[10], the MAE of the Materials Project is reported as 0.133 eV/atom. We also performed an analysis to compare the experimental formation energies of 463 materials against their corresponding formation energies from OQMD, the Materials Project and JARVIS data sets available in Matminer (an open-source materials data mining toolkit)[52]. A scatter plot of the comparison of different DFT-computed data sets against the experimental observations is illustrated in Fig. 1. We find the MAEs in OQMD, Materials Project and JARVIS are 0.083 eV/atom, 0.078 eV/atom and 0.095 eV/atom, respectively, against experimental formation energies. In this paper, we will refer to this as the "discrepancy" between DFT computation and experiments, in order to distinguish it from the "error" of the ML-based predictive models built on top of DFT/experimental data sets. As DFT calculations are performed at 0 K and the experimental formation energies are typically measured at room temperature, the two formation energies could be different[10,50]. However, such a difference is very small except for the materials that undergo phase transformation between 0 K and 300 K; these elements include Ce, Na, Li, Ti, and Sn[53]. DFT databases, such as OQMD and the Materials Project, reduce this systematic error by chemical potential fitting

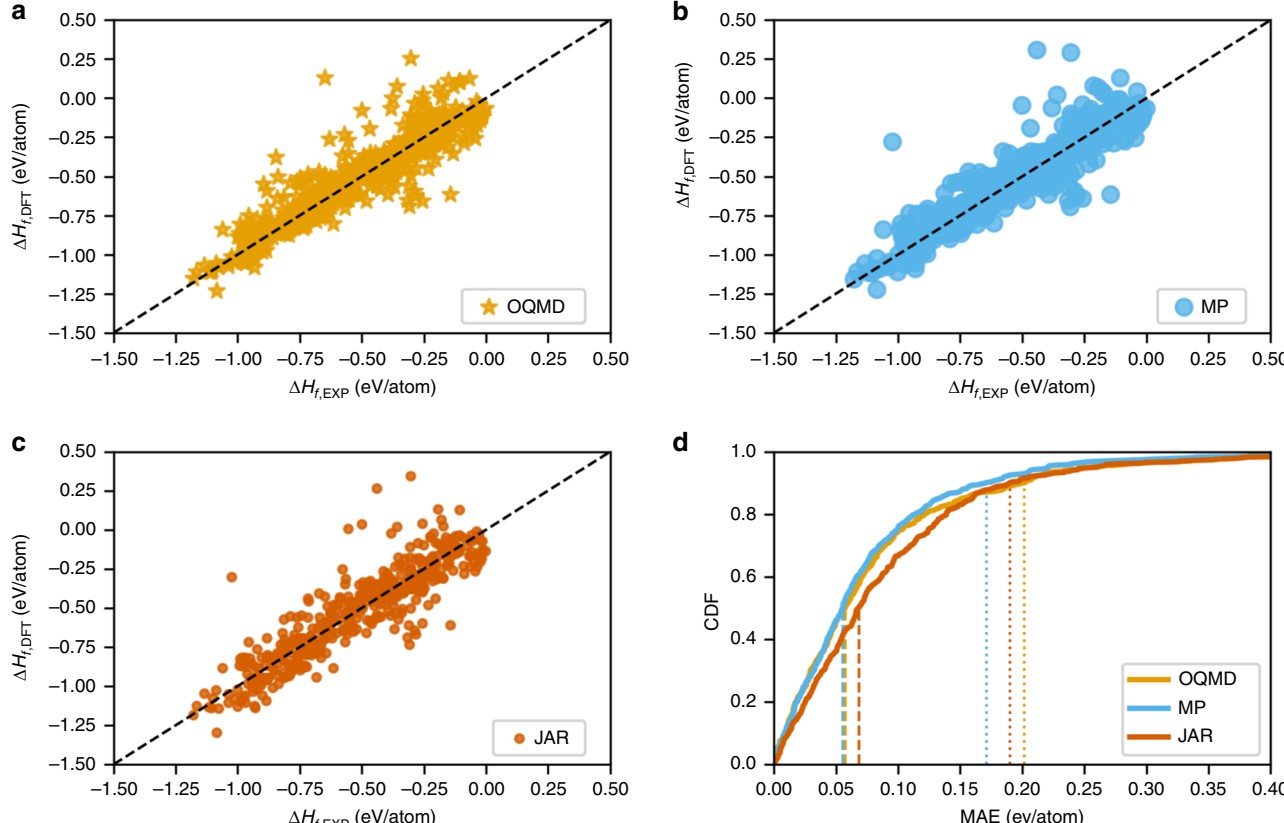

**Fig. 1** DFT-computed formation energies against the experimental observations. A comparison using scatter plots of the DFT-computed formation energies of 463 materials from **a** OQMD, **b** Materials Project (MP), and **c** JARVIS (JAR) data sets against their corresponding experimental formation energies from Matminer[52]. **d** CDF of the corresponding DFT-computation errors for the three data sets.

procedures for the constituent elements having phase transformations between 0 K and 300 K[10]. For instance, Kim et al.[50] performed a comparison between the experimental and the DFT-computed formation energy of such compounds containing constituent elements having phase transformation at low temperature, and reported an average discrepancy of ~0.1 eV/atom in both the Materials Project and OQMD; the average uncertainty of the experimental standard formation energy was one order of magnitude lower. Unlike OQMD and Materials Project, JARVIS does not apply any empirical corrections on formation energies to match experiments. As a consequence, such models trained on DFT-computed data sets automatically inherit the underlying discrepancies between the DFT computations and the experimental observations, in addition to the prediction error with respect to DFT computations used for training. The discrepancy between DFT-computation and experiments serves as the lower bound of the prediction errors that can be achieved by the ML models with respect to experiments. Owing to this issue, potential material candidates identified by such ML screening could be incorrect and disagree with intuition from domain knowledge and experiments[24,32,42].

In this work, we demonstrate that it is possible to predict material properties closer to the true experimental observations using deep learning models that can leverage the existing large DFT-computational data sets together with available experimental observations and other smaller DFT-computed data sets. Deep learning[46] enables us to perform transfer learning from large data sets to smaller data sets between similar domains. The transfer learning approach works by first training a deep neural network model on the source domain with a large data set and then, fine-tuning the trained model parameters by training on the target domain with a relatively smaller data set as shown in Fig. 2[54,55]. As the model is first trained on a large data set, it identifies a rich set of features from the input data representation, and this simplifies the task of learning features present in the smaller data set, on which the model is subsequently fine-tuned. Specifically, here we evaluate the effectiveness of the proposed approach by revisiting a commonly-studied challenge in materials informatics: predicting whether a crystal structure will be stable (formation energy) given its composition[24,32,56–58]. We leverage the recent deep neural network architecture: ElemNet[42]; ElemNet enables us to perform transfer learning from OQMD (a large data set containing DFT-computed materials properties for ~341K

materials) to two other DFT databases (JARVIS and the Materials Project) and an experimental data set containing 1963 samples from the SGTE Solid SUBstance (SSUB) database. Our results demonstrate a significant benefit from the use of deep transfer learning; in particular, the proposed approach enables us to achieve an MAE of 0.06 eV/atom against an experimental data set containing 1963 observations, which is significantly better than the mean absolute discrepancy of ~0.1 eV/atom of the DFT-computational data sets compared against experiments, and MAE of ~0.15 eV/atom of the predictive models trained from scratch (without using transfer learning) on either experimental data set or DFT-computed data sets.

## Results

**Data sets.** We use three data sets of DFT-computed properties: OQMD, the Materials Project and JARVIS, and one experimental data set. Among other properties, these databases report the composition of material compounds along with their lowest formation energy in eV/atom, hence identifying their most stable structure. OQMD contains composition and formation energies for ~341$K$ material compounds that can be either stable or unstable. We selected 11,050 stable materials from JARVIS and 23,641 stable materials from Materials Project. Note that the total number of materials in JARVIS and Materials Project is on the order of 30,000 and 70,000, respectively. However, for the present work, only materials present on the convex hull (energy above convex hull = 0) were selected. In the case of material compounds with multiple crystal structures, the minimum formation energy for the given material composition is used, as it represents the most stable crystal structure. For the experimental data set, we use the experimental formation energy from the SGTE Solid SUBstance (SSUB) database; they are collected by international scientists[59] and contain a single value of the experimental formation enthalpy, which should represent the average of formation enthalpy observed during multiple experiments, and do not contain error bars. It is curated and used by Kirklin et al.[10] in their study of assessing the accuracy of DFT formation energies in OQMD. It is composed of 1,963 formation energies at 298.15 K, and contains many oxides, nitrides, hydrides, halides, and some intermetallics, all being stable compounds.

**Training from scratch.** First, we discuss our results when training ElemNet model architecture on each data set from scratch. Although training from scratch, the model parameters are initialized randomly from a uniform distribution. As the model parameters are initialized randomly, all the features are learned from the input training data. The input vector contains the elemental fractions normalized to one, and the regression output gives the formation energy. The models learn to capture the required chemistry from the input training data. We report the results of a 10-fold cross-validation (except OQMD) performed on the four data sets in Table 1 (for OQMD, we used a 9:1 random split into train and test (validation) sets for this analysis, and the same model is used 10 times to get the predictions on the test set since the model predictions changes for same input owing to use of Dropout[69]). We also report performance of our models on a separate holdout test set using two different training:test set splits in Table 2. For holdout test, we split the data sets into training and test sets in the ratio of 9:1 and 8:2 and train ElemNet model architecture on the training sets using a 10-fold cross-validation, and report the performance of the best model from the 10-fold cross-validation on the holdout test set. Our results demonstrate that the size of training data set has a significant impact on the model performance, which is in agreement with

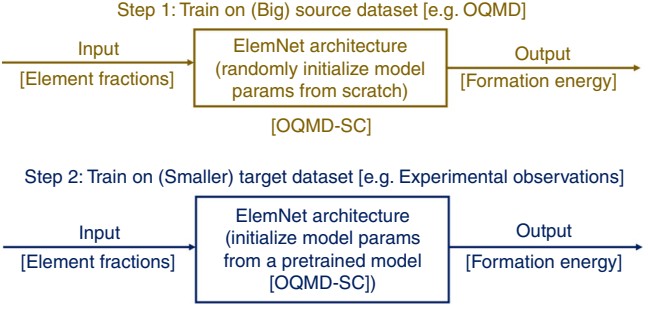

**Fig. 2** Proposed approach of deep transfer learning. First, a deep neural network architecture (ElemNet) is trained from scratch, by initializing model parameters randomly from a uniform distribution, on a big DFT-computed source data set (OQMD). As this model is trained from scratch on OQMD, we refer to this as OQMD-SC model. Next, the same architecture (ElemNet) is trained on smaller target data sets, such as the experimental data set, using transfer learning. Here, the model parameters are initialized with that of OQMD-SC, and then fine-tuned using the corresponding target dataset.

**Table 1 Performance of the ElemNet models from 10-fold cross-validation in MAE (eV/atom).**

| Data set | Size | Scratch [SC] | OQMD-SC | Transfer Learning [TL] |
|---|---|---|---|---|
| OQMD | 341, 000 | 0.0417 ± 0.0000 | – | – |
| JARVIS | 11, 050 | 0.0546 ± 0.0019 | 0.0821 ± 0.0000 | 0.0311 ± 0.0012 |
| Materials project | 23,641 | 0.0326 ± 0.0009 | 0.1084 ± 0.0000 | 0.0248 ± 0.0006 |
| Experimental | 1963 | 0.1299 ± 0.0136 | 0.1354 ± 0.0000 | 0.0642 ± 0.0061 |

**Table 2 Holdout test set performance of the ElemNet models in MAE (eV/atom).**

| Data set | Size | Train:test split ratio | Scratch [SC] | Transfer learning [TL] |
|---|---|---|---|---|
| OQMD | 341, 000 | 8:2 | 0.0471 | — |
| OQMD | 341, 000 | 9:1 | 0.0437 | — |
| JARVIS | 11, 050 | 8:2 | 0.0593 | 0.0324 |
| JARVIS | 11, 050 | 9:1 | 0.0568 | 0.0312 |
| Materials project | 23, 641 | 8:2 | 0.0347 | 0.0251 |
| Materials project | 23, 641 | 9:1 | 0.0327 | 0.0247 |
| Experimental | 1963 | 8:2 | 0.1388 | 0.0660 |
| Experimental | 1963 | 9:1 | 0.1460 | 0.0608 |

similar analyses from past studies[33,42]. Despite the smaller training data set size, the ElemNet model trained using the Materials Project has slightly better performance compared with the models trained using OQMD. This may be attributed to the inherent formation energy data in Materials Project for which several empirical fittings were applied. The impact of training data set is most evident in the case of the experimental data set, where the training data for each fold of the 10-fold cross-validation contains only ~1767 observations and each test (validation) set contains ~196 samples. The higher error in the case of the experimental data set is owing to its limited size and clearly illustrates the impact of the training data size on the performance of predictive models.

**Prediction using OQMD-SC model**. As OQMD is the largest data set used for training our models, we evaluated the ElemNet model trained on OQMD from scratch for making predictions on different data sets. We refer to this as the OQMD-SC. As shown in Table 1, we observe that although the OQMD-SC model has a low prediction error with an MAE of 0.0417 eV/atom against OQMD, it exhibits significantly higher error when evaluated against other data sets, regardless of whether they are DFT-computed or experimental. Although JARVIS, the Materials Project and OQMD are all DFT-computed data sets, they differ in their underlying approach for DFT computations. Note that the OQMD-SC model is trained using only OQMD, our goal in this evaluation is to illustrate the underlying difference in different DFT data sets and the discrepancy between OQMD and the experimental observations. When the OQMD-SC model is evaluated against JARVIS and the Materials Project, which are different from the training data set OQMD, the underlying difference in DFT computations between OQMD and the test data sets becomes obvious. This problem is exacerbated when the OQMD-SC model is evaluated on the experimental observations. As the DFT computations for the formation energy in the QOMD have a significant discrepancy (an MAE of ~0.1 eV/atom) against experimental observations, this adds up with the prediction error of the OQMD-SC model against the OQMD data set itself. If we compare the prediction errors using the OQMD-SC model on different data sets against the error of the models trained from scratch on them, we find that prediction errors are in the same

order of magnitude. The evaluation error for the Materials Project data set using OQMD-SC model is three times greater compared with the ElemNet model trained from scratch using the Materials Project. Since the empirical shifts applied in the Materials Project are not performed for OQMD, the OQMD-SC model cannot learn about them and performs poorly when evaluated on the Materials Project data set (which is different from the training data set—OQMD). Especially, in the case of the experimental data set, where the training sets in the 10-fold cross-validation contains only ~1770 compositions, the prediction error of the OQMD-SC model is very close to the model trained from scratch using the experimental data set. Such observations suggest the research question of whether using an existing model trained on large DFT-computed data sets is better than using a prediction model trained from scratch on relatively smaller data sets such as ones from experimental observations containing ~1000s samples.

**Impact of transfer learning**. As the prediction error of both the model trained from scratch on the experimental data set and the OQMD-SC model (which is trained from scratch on largest DFT-comptued data set—OQMD) against the experimental observations is poor, we decided to leverage the concept of deep transfer learning as it enables to transfer the feature representations learned for a particular predictive modeling task from a big source data set to other smaller target data sets in similar domains. For the task of transfer learning, we chose the OQMD-SC model, which is trained from scratch on OQMD using a 9:1 random split for training and the test (validation) sets. We chose the OQMD-SC model owing to two reasons. First, OQMD-SC model is trained on OQMD, which is the largest data set in our study, containing ~341 K samples. Second, the OQMD-SC model learns the required physical and chemical interactions and similarities between different elements better than other models trained from scratch, which is again owing to the large data set used for training (more on this later). The use of transfer learning helps us in leveraging these chemical and physical interactions and similarities between elements learned by the OQMD-SC model in training models for the other relatively smaller data sets. Unlike in the case of training from scratch, where the model parameters are initialized randomly, here the model parameters are initialized using the ones from the OQMD-SC model. Next, they are fine-tuned during the new training process, to learn the data representation from the smaller target data set.

We find that the prediction error significantly drops after using transfer learning from OQMD-SC model. As seen in Tables 1 and 2, the prediction error for the experimental data model almost halves. Interestingly, the error of the model trained using transfer learning from OQMD-SC model on JARVIS and the Materials Project achieves even smaller error than that of the prediction error of the OQMD-SC model itself against the OQMD data set. Since the JARVIS and Materials Project data sets are larger than the experimental data set, we observe better performance for JARVIS and Materials Project. The use of transfer learning is very effective in the case of the models trained using experimental observations. We find that the use of transfer learning from the OQMD-SC model moves the predictions closer to the true experimental

observations. The prediction error of the model trained on the experimental data set using transfer learning from OQMD-SC model is also comparable to the prediction error of the OQMD-SC model itself against the OQMD data set. We expect the benefit of using deep transfer learning to improve with the increase in the availability of experimental observations for fine-tuning (as discussed next). We believe that an MAE of 0.06 eV/atom by a prediction model against experimental observations is a remarkable feat as this is comparable to and slightly better than the existing discrepancy of DFT computations themselves against experimental observations[10].

**Impact of training data size on transfer learning.** The success of deep learning in many applications is mostly attributed to the availability of large training data sets, which has discouraged many researchers in the scientific community having access to only small data sets from leveraging deep learning in their research. In our previous work[42], we demonstrated how deep learning can be used even with small data sets (in the order of 1000s) to build more robust predictive models than the ones using traditional ML approaches like random forest. Here, we demonstrate how transfer learning can be leveraged even if the target data set is very small (in the order of 100s). We demonstrate this for the experimental data set by fixing the test (validation) set and changing the size of the training data set from 10% to 100% with an increment of 10%, for each fold in the 10-fold cross-validation. We trained the ElemNet model from scratch—EXP-SC, and also using transfer learning from OQMD-SC model—EXP-TL, on training data with varying size, as illustrated in Fig. 3. For EXP-SC, we observe a large impact of the training data set size as the MAE decreased from 0.474 ev/atom to 0.124 ev/atom as the training data size increased from 10% to 100%. However, the impact of training data set size is significantly lower in the case of transfer learning in the case of EXP-TL; the MAE changes gradually from 0.108 ev/atom to

0.064 ev/atom, as the training data size changes from 10% to 100%. This illustrates that the proposed approach of deep transfer learning can be leveraged even in the case of significantly smaller data sets having ~100s of samples for fine-tuning provided there exists a bigger source data set for transfer learning.

**Prediction error analysis.** Next, we analyzed the distribution of prediction error of all ElemNet models: the model trained from scratch (denoted by EXP-SC, JAR-SC, MP-SC), and the model trained using transfer learning from OQMD-SC model (denoted by EXP-TL, JAR-TL, MP-TL). Figure 4 illustrates the scatter plot and cumulative distribution function (CDF) of the ElemNet models trained from scratch and using transfer learning on different data sets; they contain the test predictions gathered using 10-fold cross-validation in different cases. We find that the use of transfer learning leads to significant improvement in the prediction of formation energy; the predicted values move closer to the DFT-computed or the experimental values. The benefit of the use of transfer learning is most significant in the case of experimental data; the predicted formation energies are mostly concentrated along the diagonal (hence, closer to the values from actual experimental observations). A glimpse of the CDF of the model trained using experimental data shows the same benefit in terms of percentiles; both the 50th and 90th percentiles of prediction error reduced by almost half. We observe a similar trend in case of JARVIS and Materials Project; although the distributions look similar, there is a clear reduction in prediction error as predicted values become more concentrated along the diagonal of the scatter plot in both cases. The third row in Fig. 4 illustrates the scatter plot and CDF of the OQMD-SC model against a test set containing 34,145 materials from the OQMD. Although the scatter plot appears to have a widespread in the prediction error, most of the predictions are very close to the diagonal. This is evident from the CDF plot, which illustrates that the 50th percentile error is ~0.015 eV/atom and the 90th percentile error is

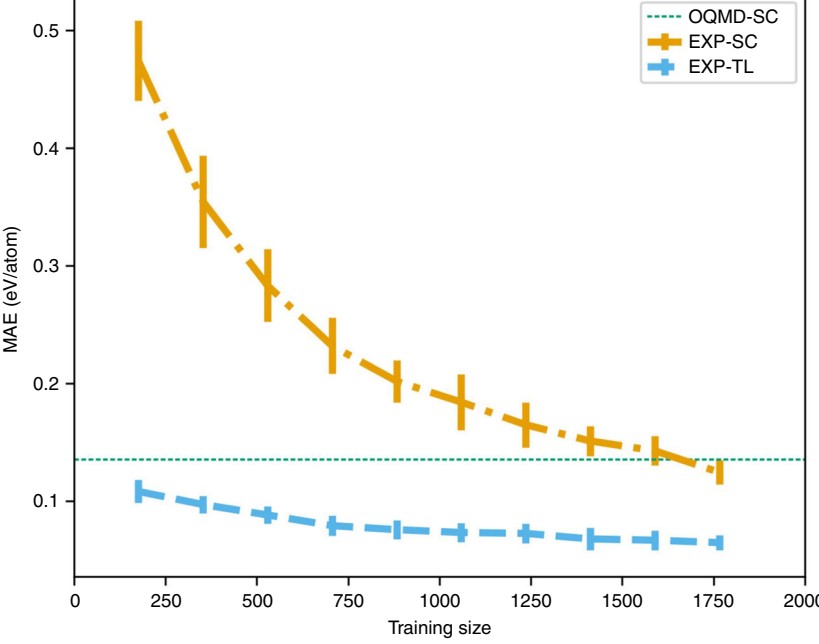

**Fig. 3** Impact of training size on model performance. The models are trained on the experimental data set and the results are aggregated from a 10-fold cross-validation (mean and standard deviation). First, we split the complete data set randomly into training and test (validation) set in the ratio of 9:1. Next, we fixed the test (validation) set and changed the size of the training set from 10% to 100%. OQMD-SC represents the model trained from scratch on OQMD data set, EXP-SC represents the model trained from scratch on the experimental data set, and EXP-TL represents the model built on experimental data set using transfer learning from the OQMD-SC model.

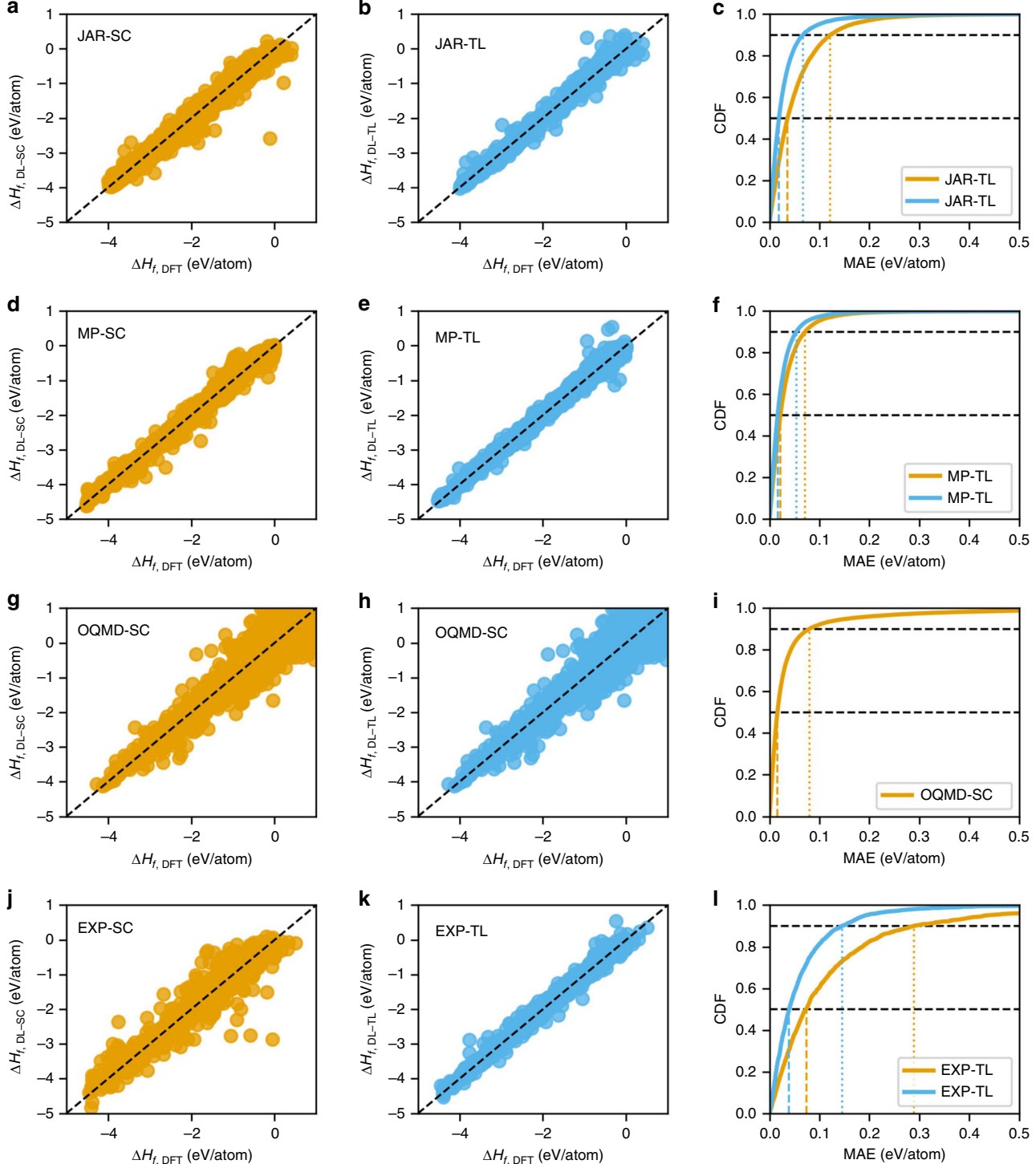

**Fig. 4** Prediction error analysis. For OQMD-SC, ElemNet model is trained from scratch using a 9:1 random split into training and test (validation) set of OQMD; here, we show the predictions on the test set. For other (smaller) data sets, we aggregate the predictions on the test (validation) sets from each split of the 10-fold cross-validation. The four rows represent the four data sets: **a–c** JARVIS (JAR), **d–f** Materials Project (MP), **g–i** OQMD, and **j–l** the experimental observations (EXP); first **a**, **d**, **g**, and **j** and second **b**, **e** and **k** (except h) columns of each row show the predictions using the model trained on the particular data set from scratch (SC) and using transfer learning (TL), respectively, the third column **c**, **f**, **i**, and **l** shows the corresponding CDF of the prediction errors using models trained from scratch (SC) and using transfer learning (TL).

~0.08 eV/atom. Hence, the OQMD-SC model predicts the formation energy of most of the compounds with high precision when compared against OQMD itself. However, OQMD-SC model has significantly worse error distribution when compared against other three data sets—broader spread in the scatter plot

and lower slopes for the CDF curves (Supplementary Fig. 1), which illustrates that although the OQMD-SC model is trained on the big DFT-computed OQMD data set, it does not always make robust predictions against data sets computed/collected using other techniques. A thorough analysis of the input elements

| Table 3 Performance of ElemNet models on the experimental data in MAE (eV/atom). | | | |
|---|---|---|---|
| Training data set | Test data set | Scratch [SC] | Transfer learning [TL] |
| OQMD | Experimental | 0.1354 ± 0.0000 | – |
| JARVIS | Experimental | 0.1911 ± 0.0042 | 0.1487 ± 0.0027 |
| Materials project | Experimental | 0.1619 ± 0.0020 | 0.1613 ± 0.0016 |
| Experimental | Experimental | 0.1299 ± 0.0136 | 0.0642 ± 0.0061 |

present in the set of compounds having more than 98th percentile error is available in the Supplementary Discussion.

**Performance on experimental data**. Next, we analyze the performance of the prediction models trained on different DFT-computed data sets (both trained from scratch and with transfer learning from the OQMD-SC model), by evaluating their performance on the experimental observations containing 1963 samples. The performance of different models on the experimental data set is shown in Table 3. For models trained on experimental data, we report the performance on test (validation) sets from the 10-fold cross-validation. For JARVIS and the Materials Project, we report the mean and standard deviation of the predictions using 10 different models from the 10-fold cross-validation. For OQMD, we use one OQMD-SC model 10 times since use of Dropout[69] results in different predictions for same input. As we can observe from these results, the performance of all the models trained on DFT-computed data sets is significantly worse compared with their performance against unseen test sets from the data set on which they are trained (Table 1). There is a minor impact of the use of transfer learning for the models trained on the JARVIS and Materials Project data set. Among all the models trained using DFT-computed data sets, the OQMD-SC model has the lowest discrepancy which is comparable to the prediction error of model trained on experimental data set from scratch. The performance of OQMD-SC model re-emphasizes the impact of training data size, which enables the model to automatically capture the physical and chemical interactions from the input data representation that is essential for making correct predictions. The error in predictions using different models are at least double than that of the model trained on the experimental data set using transfer learning from the OQMD-SC model. Our observations demonstrate the need to leverage DFT-computed data sets with experimental data sets to build robust prediction models, which can make predictions closer to true experimental observations, thereby questioning and providing an alternative to the current practice of using predictive models built using DFT-computed data sets alone.

Figure 5 illustrates the scatter plot of the predicted values against the true experimental values and CDF of the corresponding errors. If we look at the prediction results using the OQMD-SC model in Fig. 5, the predictions are less concentrated on the diagonal of the scatter plot; the 50th percentile error is 0.1 eV/atom and the 90th percentile error is 0.28 eV/atom. This is significantly worse than the test error of OQMD-SC model on OQMD itself (MAE of 0.04 eV/atom in Table 1) and the discrepancy of the DFT computations for OQMD against experimental values (0.1 eV/atom[10]). This illustrates the high deviation of the OQMD-SC model in the predicted values against the true experimental observations. The improvement owing to transfer learning in the prediction error distribution is negligible for the models trained using JARVIS and Materials Project data sets. This again illustrates the inefficacy of using a model trained using DFT-computed data sets alone, since they will have high

prediction error against experimental observations owing to the inherent discrepancy of the DFT computation itself against experimental observations.

**Activation analysis**. Next, to understand the impact of transfer learning on the performance of models trained using different data sets, we analyzed the activations from different layers of ElemNet architecture to visualize the physical and chemical interactions and similarities captured by the model. We performed two kinds of analysis for two different classification tasks using two different data sets. The first analysis involved taking the activations from each layer of different models and apply principal component analysis (PCA) for dimensionality reduction; since the number of activations varies from 1024 in the first hidden layer to 32 in the penultimate layer, we use PCA to get first two principal components and scale them in the range of [0,1] for ease of visualization using a scatter plot. The second analysis involved taking the activations from each hidden layer without applying PCA and training a Logistic Regression for classification using a random split of training and test set in the ratio of 9:1. We analyze the activations to see how well they can be used to perform three classification tasks—magnetic vs non-magnetic (1 vs 0) from JARVIS, insulator vs metallic (1 vs 0) from JARVIS, and insulator vs metallic (1 vs 0) from Materials Project.

Figure 6 demonstrates the scatter plot and ROC (Receiver Operating Characteristics) curves of the Logistic Regression model trained using activations from the first hidden layer of the ElemNet model trained from scratch and using transfer learning on different data sets. Logistic Regression is a statistical model based on using a logistic function to model the binary dependent variable for binary classification problems[60,61]. A ROC curve is generated by plotting the true positive rate (TPR) against the false positive rate (FPR) at a varying threshold, and the area under the curve (AUC) of a ROC curve represents the performance measurement for the binary classification problem[62]. Higher the AUC of a ROC, better is the model at distinguishing between the binary classes. The distinction of magnetic vs non-magnetic materials is evident from the visualization using the scatter plot of the first two PCA components of the activations of the same hidden layer in Fig. 6. In the case of the OQMD-SC model, we find that the distinction between the two classes is more distinguished, which agrees with the fact that the ElemNet model trained on OQMD data set captures the physical and chemical interactions between different elements automatically[42]. From the scatter plot of the first two components of the PCA analysis, we find that other than the OQMD-SC model, other models trained from scratch hardly capture the distinction between magnetic and non-magnetic class (1 vs 0) from the training data set, owing to their relatively small size used for training (first row of Fig. 6). When using transfer learning, we find that this ability to distinguish between magnetic and non-magnetic is passed to the fine-tuned models, thereby enhancing the prediction performance of the models trained using transfer learning from the ElemNet-QOMD model. Although there is no clear boundary between the magnetic vs non-magnetic materials in the scatter plot, the magnetic materials are concentrated towards the lower part of the scatter plot for the models trained using transfer learning.

This enhancement in the ability to distinguish between magnetic and non-magnetic materials becomes more evident if we look at the ROC curve of the Logistic Regression model trained using the actual activations from the same layer. As shown in Fig. 6, the Logistic Regression models trained using activations from the model trained using transfer learning from OQMD-SC model exhibit a significant difference in the AUC of

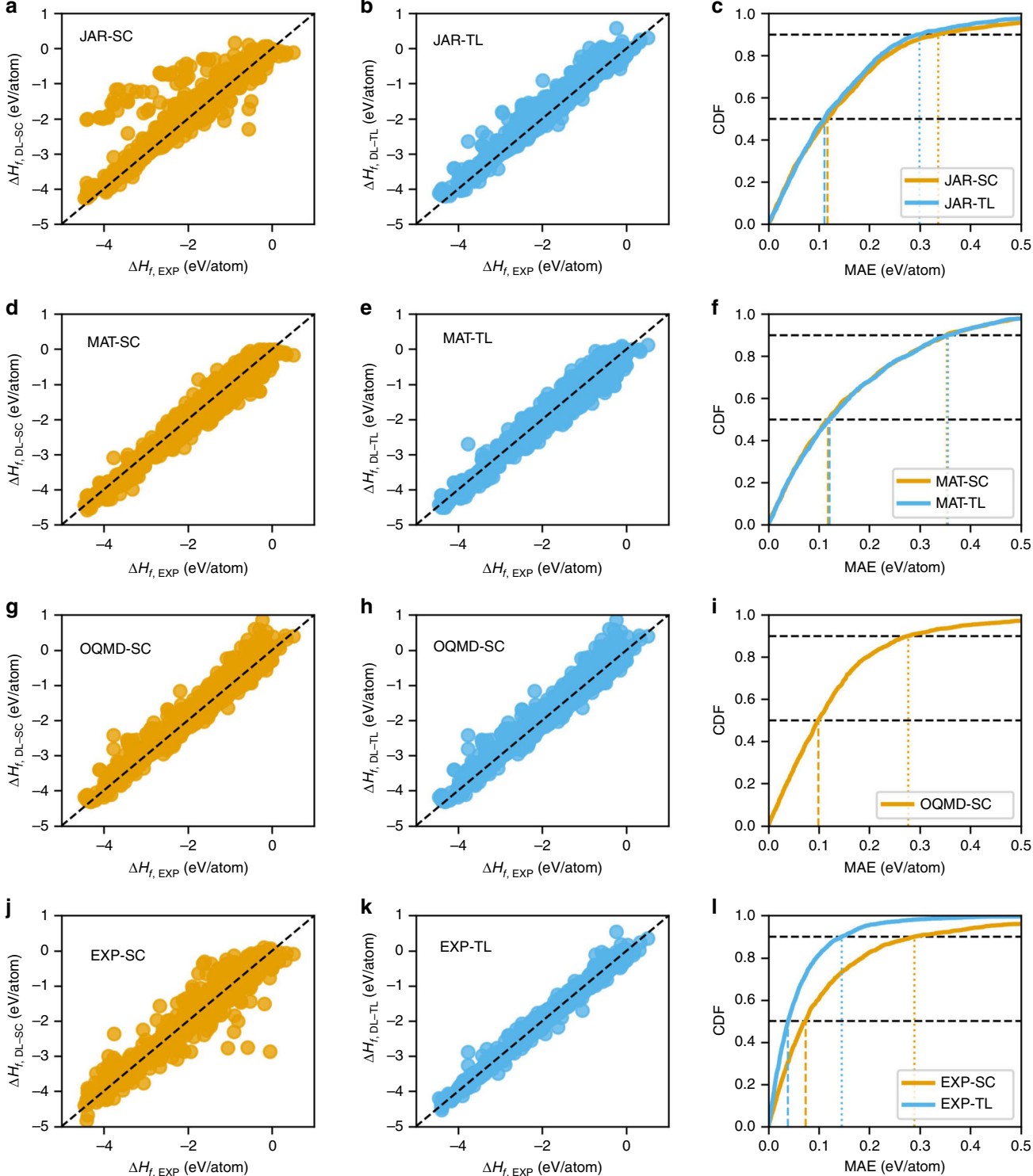

**Fig. 5** Prediction error analysis on the experimental data set. The experimental data set contains 1,963 observations. For the models trained using experimental data set, the predictions are aggregated on test (validation) sets from each split in the 10-fold cross-validation. For the models trained using JARVIS and Materials Project, since we have 10 models from the 10-fold cross-validation during training, we take the mean of their predictions for each data point in the experimental data set. For OQMD-SC, we make 10 predictions for each point in the experimental data set and take their mean. The four rows represent the four data sets: **a–c** JARVIS (JAR), **d–f** Materials Project (MP), **g–i** OQMD, and **j–l** the experimental observations (EXP); first **a**, **d**, **g**, and **j** and second **b**, **e** and **k** (except **h**) columns of each row show the predictions using the model trained on the particular data set from scratch (SC) and using transfer learning (TL) respectively; the third column **c**, **f**, **i**, and **l** shows the corresponding CDF of the prediction errors using models trained from scratch (SC) and using transfer learning (TL).

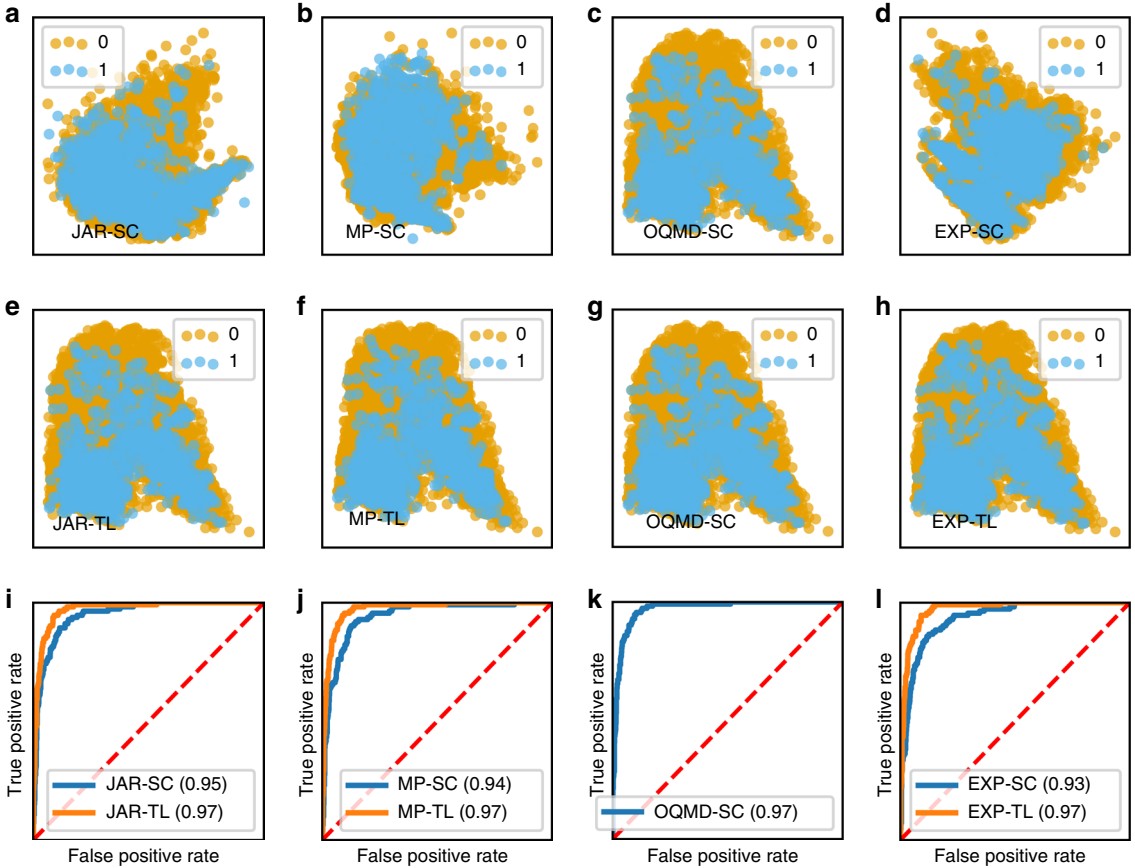

**Fig. 6** Activation analysis to understand the impact of transfer learning. Here, we analyze the activations from the first hidden layer of the ElemNet architecture for understanding the impact of transfer learning on the model's capability to automatically learn to distinguish between the magnetic vs non-magnetic class (1 and 0) from JARVIS data set. The four columns represent the models trained using four different data sets: **a**, **e**, and **i** using JARVIS (JAR), **b**, **f**, and **j** using Materials Project (MP), **c**, **g**, and **k** using OQMD and **d**, **h**, and **l** using the experimental observations (EXP); the first **a–d** and second **e–h** (except g) rows represent scatter plots demonstrating the first two principal components of the activations using principal component analysis (PCA) technique from the models trained from scratch (SC) and using transfer learning (TL), whereas third row **i–l** represents the ROC curves from the Logistic Regression model trained using complete set of activations from the same hidden layer (the corresponding AUC values are shown in brackets) on the corresponding data sets.

the ROC curve−0.97 compared with that of ~0.93 using the activations from the model trained from scratch (except the OQMD-SC model). We observe a similar impact on the classification task to distinguish magnetic and non-magnetic materials for activations up to the first six layers. Further, we observed similar results for insulator vs metallic class for different data sets, and the analysis for JARVIS data set is available in the Supplementary Fig. 2. We also performed this task on activations of different layers for the data set from the Materials Project, and observed similar results. An interesting observation is that although the activation plots of all the different models trained from scratch look distinct, they look almost similar after the use of transfer learning from the OQMD-SC model. This illustrates that the knowledge of chemical and physical interactions and similarities between different elements transferred from the OQMD-SC model dominates even after the models are fine-tuned using the target data sets; this is because data representation learned from OQMD is very rich compared with the limited representation present in the relatively smaller training data sets from JARVIS, the Materials Project and the experimental observations.

## Discussion

In this work, we demonstrated the benefit of leveraging both DFT computations and experimental observations to build more robust prediction models whose predictions are closer to the experimental observations compared with the predictive models built using only DFT-computed data sets. As we already illustrated how ElemNet can automatically capture the underlying chemistry from only elemental fractions using artificial intelligence (deep learning) and perform better than the traditional ML approach in our previous work[42], here we focused on using the deep neural network architecture of ElemNet for deep transfer learning of the chemistry learned from large data sets to smaller data sets using DFT or experimental observations; the comparison of ElemNet against traditional ML approaches for all data sets is available in the Supplementary Table 1. Our analysis of the prediction models based on different DFT-computed and experimental data sets illuminates the fundamental problem of building prediction models using DFT-computed data sets. Prediction models built using only the DFT-computed values exhibit high prediction errors against the experimental values; this results from the inherent discrepancy of DFT computations against the experimental observations themselves, in addition to the error of the model against the DFT-computed values used for its training. We expect the proposed approach to perform better with the increasing availability of DFT computations (for source data set) as well as an increase in the experimental observations for fine-tuning.

We have shown the application of deep transfer learning in predicting formation energy of materials (and hence, the stability

of materials) such that they are closer to experimental observations, which in turn, can be used for performing more robust combinatorial screening for hypothetical materials candidates for new materials discovery and design[24,42]. Formation energy is an extremely important material property since it is required to predict compound stability, generate phase diagrams, calculate reaction enthalpies and voltages, and determine many other important properties. Note that while formation energy is so ubiquitous, DFT calculations allow prediction of many other properties (such as bandgap energy, volume, energy above the convex hull, elasticity, magnetization moment), which are very expensive to measure experimentally. The presented approach can be leveraged for predicting many other such materials properties where we have large computational data sets (such as using DFT), but small ground truth (experimental observations), a scenario that is very common in materials science; some examples being predicting bandgap energies of certain classes of crystals[32,63,64], thermal conductivity, thermal expansion coefficients, Seebeck coefficient of thermal compounds[65,66], mechanical properties of metal alloys[36,63], magnetic properties of materials[25], and so on, for various types of applications in materials design. DFT databases are in the order of $10^4$, however, the computationally hypothetical materials are in the order of $10^{10}$, that is where ML models can be extremely valuable for the pre-screening process[24,42]. As long as the source data set for transfer learning contains a diverse range of chemistry and the target data set contains compounds having similar chemistry (a subset of elements or features present in the source data set for transfer learning), we expect the proposed method to work well. The presented approach can also be leveraged for building more robust predictive systems for other scientific domains where the amount of experimental observations and ground truth is not sufficient to train a ML model on its own, but there exists a large set of computational/simulation data set from the same domain for transfer learning.

## Methods

**Data cleaning**. The input data are composed of fixed size vectors containing raw elemental compositions as the input and formation enthalpy in eV/atom as the output labels. The input vector is composed of non-zero values for all the elements present in the compound and zero values for others; the composition fractions are normalized to one. We perform two stages of data cleaning to remove single elements and outliers. The single elements are removed since their formation energy is zero. The samples with formation energy outside of $\pm 5\sigma$ ($\sigma$ is the standard deviation in the training set) are removed. Further, the elements not appearing in the training data sets after cleaning are removed from the input attribute set. Out of 118 elements in the periodic table, our data set contains the following 86 elements—[H, Li, Be, B, C, N, O, F, Na, Mg, Al, Si, P, S, Cl, K, Ca, Sc, Ti, V, Cr, Mn, Fe, Co, Ni, Cu, Zn, Ga, Ge, As, Se, Br, Kr, Rb, Sr, Y, Zr, Nb, Mo, Tc, Ru, Rh, Pd, Ag, Cd, In, Sn, Sb, Te, I, Xe, Cs, Ba, La, Ce, Pr, Nd, Pm, Sm, Eu, Gd, Tb, Dy, Ho, Er, Tm, Yb, Lu, Hf, Ta, W, Re, Os, Ir, Pt, Au, Hg, Tl, Pb, Bi, Ac, Th, Pa, U, Np, and Pu].

**Experimental settings and tools used**. We have used the ElemNet[42] model architecture shown in Table 4 implemented using Python and TensorFlow[67] framework. ElemNet is a 17-layered fully connected deep neural network architecture that is designed to predict the formation energy from elemental fractions without any manual feature engineering[42]. The input for ElemNet is composed of a set of 86 elements in our data set, from Hydrogen to Plutonium except for Helium, Neon, Argon, Polonium, Astatine, Radon, Francium, and Radium. These 86 elements form the materials in most of the current DFT-computed data sets such as OQMD, JARVIS, and the Materials Project. ElemNet model is trained on each data set with/without using transfer learning using 10-fold cross-validation except when training from scratch on OQMD; in the case of OQMD, ElemNet model is trained using a 9:1 random split into train and test (validation) sets, this is referred as OQMD-SC. OQMD-SC is used for transfer learning in this work. We train for 1000 epochs with a learning rate of 0.0001 and minibatch size of 32 using Adam[68] optimizer. A patience of 200 minibatch iterations is used to avoid overfitting to the training data set; if there is no improvement in validation error for 200 minibatch iterations, the training is stopped. Dropout[69] layers are leveraged to prevent overfitting and they are not counted as a separate layer. We used ReLU[70] as the activation function. We have used the Matplotlib library in Python to plot the

**Table 4 ElemNet model architecture used for training different models.**

| Layer types | No. of units | Activation | Layer positions |
| --- | --- | --- | --- |
| Fully connected layer | 1024 | ReLU | First to 4th |
| Dropout (0.8) | 1024 | | After 4th |
| Fully connected layer | 512 | ReLU | 5th to 7th |
| Dropout (0.9) | 512 | | After 7th |
| Fully connected layer | 256 | ReLU | 8th to 10th |
| Dropout (0.7) | 256 | | After 10th |
| Fully connected layer | 128 | ReLU | 11th to 13th |
| Dropout (0.8) | 128 | | After 13th |
| Fully connected layer | 64 | ReLU | 14th to 15th |
| Fully connected layer | 32 | ReLU | 16th |
| Fully connected layer | 1 | Linear | 17th |

figures used in this manuscript. All the models are trained and tested using Titan X GPUs on NVIDIA DIGITS DevBox. The training curves of the ElemNet models trained from scratch and using transfer learning on the experimental data set are available in Supplementary Fig. 3.

## Data availability

No data sets were generated during current study. All the data sets used in the current study are available from their corresponding public repositories—OQMD (http://oqmd.org), Materials Project (https://materialsproject.org), JARVIS (https://jarvis.nist.gov), and experimental observations (https://github.com/wolverton-research-group/qmpy/blob/master/qmpy/data/thermodata/ssub.dat).

## Code availability

All the codes required to train the ElemNet model used in this study is available at https://github.com/dipendra009/ElemNet.

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

## Acknowledgements

This work was performed under the following financial assistance award 70NANB19H005 from US Department of Commerce, National Institute of Standards and Technology as part of the Center for Hierarchical Materials Design (CHiMaD). Partial support is also acknowledged from DOE awards DE-SC0014330, DE-SC0019358.

## Author contributions

D.J. designed and carried out the implementation and experiments for the deep learning model under the guidance of A.A., A.C., and W.L.. K.C., F.T., and C.C. provided the necessary domain expertize for this work. All authors discussed the results and contributed to the writing of the manuscript.

## Competing interests

The authors declare that they have no competing interests.
