## [Peer Review File · Nature Communications]

Reviewers' comments:

Reviewer #1 (Remarks to the Author):

The authors report on machine-learning (ML) prediction models of materials properties focusing on phase formation energy and utilizing some of the existing DFT and experimental databases. They demonstrate that the learning transfer strategies can significantly improve the robustness of the ML predictions. This information can be very helpful for the materials design where properties of materials can be screened utilizing the rich information contained in the existing DFT/experimental databases. The approach proposed by the authors is very sound and the results are important enough to warrant publication in Nature Communications after the authors properly address the following comments:

1. While the DFT calculations give the phase formation energies at $T = 0$ K, the experimental formation energies are usually obtained by high-temperature measurements. This obviously creates some incompatibility between the DFT and experimental formation energies. The authors should point to this difference and estimate (or at least discuss) its potential contribution to the prediction errors. I believe this discrepancy sets a lower bound of the prediction errors that can be achieved by all ML models for the DFT->Experiment extrapolation. This factor should be taken into account, in addition to the smaller size of the experimental database. It may also limit the effectiveness of the knowledge transfer from a large DFT database (such as OQMD) to an experimental dataset.

2. More information should be provided about the network architecture (number of layers, number of nodes in the layers, etc) and thus the total number of fit parameters available for training. Was the network the same for the large (e.g. OQMD) and smaller (experiment, JARVIS) datasets? The results cannot be reproduced by others without these details.

3. What was the stopping condition during the training? In other words, how much further would the MAE go down if the authors continued the training before they reach overfitting? Would this deeper training reduce the beneficial effect of the pre-learned knowledge?

4. A couple of minor things: (1) the paper is too long for this journal, and in fact slightly overwritten for its scientific content. The same material can be presented in a more succinct manner while all details can be moved to a Supplementary Information file. (2) The abbreviation CDF is used before it is defined, but then defined twice. (3) The blue and green colors on the CDF-MAE plots are barely distinguishable. I recommend a different choice of colors.

Reviewer #2 (Remarks to the Author):

This paper is both timely and novel, and addresses the key issue of the inherent errors in DFT (or any) electronic structure calculation database and experimental reality. The authors have done a good job of analyzing their results in a reasonably thorough fashion. This should be published once the following issues are addressed.

1. The logic of the paper is misleading. The DFT-based ML methods are not designed to reproduce experiment, just to reproduce other DFT calculations not in the training data. Thus their error relative to experiment is NOT considered an error by some. All aspects of the manuscript, from abstract onwards, should be revised to reflect this, and not refer to this as an error, but rather as a difference that the current work is exploring.

In this respect, it would be useful if the authors could test their methods on lower level DFT calculations (eg LDA), to see if training with a bigger difference would still yield as good results on the experimental data (OPTIONAL).

2. No account seems to be taken of the possible errors in the experimental database. Do they even have error bars? This issue must be addressed, preferably accounting for both statistical error bars and possibly flawed data in the experimental data set. If the methodology does not allow this, their must be some explanation given for why the authors believe such errors are irrelevant and publication may depend on the plausibility of their answer.

3. While heats of formation are extremely important, DFT calculations allow prediction of many other properties, which could be very expensive to measure. This should be addressed in the manuscript.

4. Learning curves are extremely important, and must be shown when learning on the experimental data, and their significance discussed. For example, it would allow estimation of the amount of experimental data needed to achieve the same level of accuracy with the worse-performing methods. How few data can one get away with?

5. Please add lists of the worst performing cases, at least in supplemental info, for all models, stating which systems and how bad the models are. There may be patterns.

Reviewer #3 (Remarks to the Author):

This paper describes the application of a machine learning approach to be able to model properties from DFT calculations and experiments based on minimal required input. The main objective here is to train models on large DFT-derived databases and then apply the model to smaller datasets, such as derived from experiments. While this is an interesting approach, particularly given the typically small experimental data sizes, there are several concerns which would seem to limit its applicability for a Nature based journal.

- While this approach demonstrates the approach, it is unclear what future application this approach has. Where would this work be applied or what chemical spaces are there that can be explored with this. It seems the DFT databases are pretty thorough, so where we would use this next would be interesting to consider.
- On that point, the issue of the transferability of this method is unclear. It is stated a couple times that the OQMD model cannot be applied to the Materials Project. I wonder if that means that for this particular case, the model has been trained well, and can describe what is already known. But if it cannot work well on Materials Project, does that mean its applicability to different datasets is unclear?
- From all of this work, it does not seem that there is anything scientifically that has been learned. For example, that we have identified some chemistry that was not well described before but now is, or from one of the outliers learned something, or predicted behavior of new chemistries.
- More thorough comparison with prior approaches would be helpful. There are many papers coming out on machine learning applied to these databases; how much better is this method than those other examples or what can be done that couldn't otherwise?
- Also, the main benefit of this work seems to be derived from the ElemNet method --- there seems to be a lot of overlap with the authors' paper in Scientific Reports. More discussion on what has been gained would help the reader better assess this work.
- Figure 6 and the corresponding discussion are hard to understand – while we can see there is some improvement in the ElemNet-OQMD approach, it does not appear to be highly significant.

Some context for how much that difference impacts the selection of chemistries would be helpful to understand the impact of this work.

- A small question, but it is discussed that the application of the machine learning model to the smaller experimental datasets provides good results. I was wondering the diversity of chemistries in the experimental dataset – is it more limited than the DFT trained data, thus explaining why it works, or is it able to be applied to as wide of a chemical range.

Response to Reviewer Comments

Manuscript ID: NCOMMS-19-04088

Title: Enhancing Materials Property Prediction by Leveraging Computational and Experimental Data using Deep Transfer Learning

Authors: Dipendra Jha, Kamal Choudhary, Francesca Tavazza, Wei-keng Liao, Alok Choudhary, Carelyn Campbell, and Ankit Agrawal

The authors would like to thank the reviewers for their insightful comments for improving this manuscript. Below we provide point-by-point responses to the comments along with corresponding amendments made in the manuscript.

Reviewer #1 (Remarks to the Author):

The authors report on machine-learning (ML) prediction models of materials properties focusing on phase formation energy and utilizing some of the existing DFT and experimental databases. They demonstrate that the learning transfer strategies can significantly improve the robustness of the ML predictions. This information can be very helpful for the materials design where properties of materials can be screened utilizing the rich information contained in the existing DFT/experimental databases. The approach proposed by the authors is very sound and the results are important enough to warrant publication in Nature Communications after the authors properly address the following comments:

1. While the DFT calculations give the phase formation energies at $T = 0$ K, the experimental formation energies are usually obtained by high-temperature measurements. This obviously creates some incompatibility between the DFT and experimental formation energies. The authors should point to this difference and estimate (or at least discuss) its potential contribution to the prediction errors. I believe this discrepancy sets a lower bound of the prediction errors that can be achieved by all ML models for the DFT->Experiment extrapolation. This factor should be taken into account, in addition to the smaller size of the experimental database. It may also limit the effectiveness of the knowledge transfer from a large DFT database (such as OQMD) to an experimental dataset.

→ Thank you, we have added a paragraph in the introduction (first paragraph on page 4) that discusses the discrepancy resulting from the difference in temperature between DFT and experiments, how they are handled by DFT datasets such as OQMD and Materials Project [1,2,3], and the lower error bound between DFT-based ML models with respect to experiments. Transfer learning requires large source dataset (OQMD in our case) from a similar domain. We agree that if the experimental dataset becomes larger, the fine-tuning would be better resulting in smaller prediction error after transfer learning. We have added more discussion about the impact of training dataset size on the performance (last paragraph on page 8-9, and second paragraph on page 12), we have made it more clear by adding a new analysis of the impact of training data size on transfer learning (page 12-13 (including Figure 2)).

2. More information should be provided about the network architecture (number of layers, number of nodes in the layers, etc) and thus the total number of fit parameters available for training. Was the network the same for the large (e.g. OQMD) and smaller (experiment, JARVIS) datasets? The results cannot be reproduced by others without these details.

→ Thank you, we have added more details about network architecture and the hyperparameters for training to the Method section (added table 3 on page 28). The same network was used for all datasets.

3. What was the stopping condition during the training? In other words, how much further would the MAE go down if the authors continued the training before they reach overfitting? Would this deeper training reduce the beneficial effect of the pre-learned knowledge?

→ Thank you, we used early stopping with a patience of 200 minibatch iterations during training, we have added these details to Method section (table 3 on page 28). Early stopping is a method which stops the training when the validation error do not improve after a fixed number of minibatch iterations (patience); early stopping along with dropouts are used to avoid overfitting to training dataset. We are not sure what the reviewer means by ‘*deeper training*’, if it means training for longer allowing overfitting to training dataset, we expect that not to have much impact on the beneficial effect on the pre-learned knowledge for transfer learning. If they mean deeper architecture, the deeper architecture should not reduce the benefit of the pre-learned knowledge (our model architecture is deep, having 17 layers).

4. A couple of minor things: (1) the paper is too long for this journal, and in fact slightly overwritten for its scientific content. The same material can be presented in a more succinct manner while all details can be moved to a Supplementary Information file. (2) The abbreviation CDF is used before it is defined, but then defined twice. (3) The blue and green colors on the CDF-MAE plots are barely distinguishable. I recommend a different choice of colors.

→ Thank you, we have moved some details to Supplementary Information file. We corrected the use of CDF. The colors we chose were the ones suitable for color-blind people, we changed the figures to a different set of colors for color-blind people.

Reviewer #2 (Remarks to the Author):

This paper is both timely and novel, and addresses the key issue of the inherent errors in DFT (or any) electronic structure calculation database and experimental reality. The authors have done a good job of analyzing their results in a reasonably thorough fashion. This should be published once the following issues are addressed.

1. The logic of the paper is misleading. The DFT-based ML methods are not designed to reproduce experiment, just to reproduce other DFT calculations not in the training data. Thus their error relative to experiment is NOT considered an error by some. All aspects of the manuscript, from abstract onwards, should be revised to reflect this, and not refer to this as an error, but rather as a difference that the current work is exploring.

→ Thank you for your suggestion, we have revised throughout our manuscript to reflect your point (page 17, 18 (table 2), 19 (figure 6), and 25).

In this respect, it would be useful if the authors could test their methods on lower level DFT calculations (eg LDA), to see if training with a bigger difference would still yield as good results on the experimental data (OPTIONAL).

→ We thank you for the interesting comment. We would be interested in testing the effect of the DFT functional (LDA vs GGA) explicitly but, unfortunately, to our knowledge, we are not aware of any openly available LDA databases. The only comment we can make along this line of inquiry is that the DFT exchange-correlation functional is different between OQMD/Materials Project (PBE) and JARVIS (OPTB88) and no major effect is noticed in the transfer learning.

2. No account seems to be taken of the possible errors in the experimental database. Do they even have error bars? This issue must be addressed, preferably accounting for both statistical error bars and possibly flawed data in the experimental data set. If the methodology does not allow this, there must be some explanation given for why the authors believe such errors are irrelevant and publication may depend on the plausibility of their answer.

→ Thank you, we have used the experimental formation energy from the SGTE Solid SUBstance (SSUB) database; they are collected by international scientists, and contain single value of the experimental formation enthalpy. It is

curated and used by Kirklin et al. in their study about assessing the accuracy of OQMD [3], they do not contain error bars. We have added these details in the manuscript (first paragraph on page 8).

3. While heats of formation are extremely important, DFT calculations allow prediction of many other properties, which could be very expensive to measure. This should be addressed in the manuscript.

→ Thank you, we completely agree and see this type of application as one of the strengths of this approach. We mentioned the possibility of doing this in the manuscript (page 2, 5, 26). However, this work is meant to be as a proof of concept, which is why we only focused testing transfer learning on one property, and we focused more on comparing transfer learning results between DFT databases versus between DFT and experimental data. Now that we have obtained such encouraging results, we do plan to investigate the application of this methodology to other physical quantities in future works.

4. Learning curves are extremely important, and must be shown when learning on the experimental data, and their significance discussed. For example, it would allow estimation of the amount of experimental data needed to achieve the same level of accuracy with the worse-performing methods. How few data can one get away with?

→ Thank you, we have added learning curves for experimental dataset to the supplementary information. Typically, learning curves for deep learning models show the training and validation errors during training and helps in visualizing the overfitting. To avoid overfitting to training dataset, we have used a patience of 200 iterations, that is, the training stops if the validation error does not improve in last 200 minibatch iterations. However, it appears from the comment that the reviewer might be interested to know about the other type of learning curves depicting the impact of training data size with respect to error. The same is now included on page 12-13 (including figure 2). As discussed in the paper, we find that the ElemNet-OQMD (model trained from scratch on OQMD) performs similar to the ElemNet-Exp (the model trained from scratch on experimental dataset), both having MAE around 0.13 eV/atom when evaluated using experimental dataset. ElemNet-OQMD-EXP model trained using transfer learning, outperforms all ElemNet models trained from scratch, even when we used only 10% (176 samples) for fine-tuning; this illustrates the efficiency of the proposed approach towards robust and accurate predictive modeling.

5. Please add lists of the worst performing cases, at least in supplemental info, for all models, stating which systems and how bad the models are. There may be patterns.

→ Thank you for the suggestion. We added an extensive discussion of worst performing cases in the supplementary information.

Reviewer #3 (Remarks to the Author):

This paper describes the application of a machine learning approach to be able to model properties from DFT calculations and experiments based on minimal required input. The main objective here is to train models on large DFT-derived databases and then apply the model to smaller datasets, such as derived from experiments. While this is an interesting approach, particularly given the typically small experimental data sizes, there are several concerns which would seem to limit its applicability for a Nature based journal.

1. While this approach demonstrates the approach, it is unclear what future application this approach has. Where would this work be applied or what chemical spaces are there that can be explored with this. It seems the DFT databases are pretty thorough, so where we would use this next would be interesting to consider.

→ Thank you, the presented approach have numerous future applications. We have shown the application of deep transfer learning in predicting formation energy of materials (and hence, stability of materials) such that they are closer to experimental observations, which in turn, can be used for performing more robust combinatorial screening for hypothetical materials candidates for new materials design and discovery. The presented approach can be used in predicting many other materials properties where we have large computational datasets (such as using DFT), but

small ground truth (experimental observations), a scenario which is very common in materials science; some examples being predicting band gap energies of certain classes of crystals, mechanical properties of metal alloys, magnetic properties of materials, for various types of applications in materials design. DFT databases are generally in the order of 10^4 , however, the computationally hypothetical materials are in the order of 10^{10} ; that is where machine learning models are extremely valuable for pre-screening process. As long as the source dataset for transfer learning contains a diverse range of chemistry and the target dataset contains the compounds having similar chemistry (a subset of elements or features present in the source dataset for transfer learning), we expect the proposed method to work well. The presented approach can also be leveraged in building more robust predictive systems for other scientific domains where the experimental observations and ground truth is not sufficient to train a machine learning model on its own, but there exists a large set of computational/simulation dataset from the same domain for transfer learning. We have added this to the discussion section (page 26).

2. On that point, the issue of the transferability of this method is unclear. It is stated a couple times that the OQMD model cannot be applied to the Materials Project. I wonder if that means that for this particular case, the model has been trained well, and can describe what is already known. But if it cannot work well on Materials Project, does that mean its applicability to different datasets is unclear?

→ Thank you, we are extremely sorry for the ambiguity rising from the section on evaluation performance of ElemNet-OQMD on different datasets. Since the ElemNet-OQMD is trained on OQMD which follows slightly different DFT computation methodology than Materials Project, we observe higher prediction error if we evaluate the ElemNet-OQMD model on other DFT datasets (*different from OQMD dataset on which the model is trained*) - JARVIS and Materials Project. This evaluation is completely different from the concept of deep transfer learning where we illustrate how we can use transfer learning from large dataset (ElemNet-OQMD, a model trained on OQMD) to build more robust predictive models on smaller datasets from Materials Project, JARVIS and experimental observations. We have made this point more clear by editing that evaluation section for ElemNet-OQMD (page 10).

3. From all of this work, it does not seem that there is anything scientifically that has been learned. For example, that we have identified some chemistry that was not well described before but now is, or from one of the outliers learned something, or predicted behavior of new chemistries.

→ Thank you, our focus is not to present what new things we learned in chemistry, rather our focus is on presenting a better predictive modeling approach that is more closer to experimental observations comparable to the discrepancy of DFT itself against experimental observations. To that end, we have already shown how the predicted formation energies are closer to experimental observations than ones using existing predictive modeling approach based solely on DFT computations. The activation analysis demonstrates how the ElemNet-OQMD model automatically learns the underlying chemistry which is transferred to build a more robust and accurate predictive models on the smaller dataset. More details on how the ElemNet model learns the underlying chemistry is already presented in our previous paper [4] and hence, we do not repeat it here.

4. More thorough comparison with prior approaches would be helpful. There are many papers coming out on machine learning applied to these databases; how much better is this method than those other examples or what can be done that couldn't otherwise?

→ Thank you, we have added a new comparison of the proposed approach against existing traditional machine learning approaches to Supplementary Information for interested readers; the state-of-the-art traditional machine learning approach for the prediction of formation energy given material composition - Random Forest, has a MAE of 0.1227 eV/atom which is around 100% greater than that of 0.0642 eV/atom using the proposed approach for the experimental dataset. The traditional machine learning approaches such as Random Forest, SVMs, are not suitable for transfer learning from existing large DFT datasets (such as OQMD) to other smaller datasets (such as experimental dataset), which is possible using the proposed approach.

5. Also, the main benefit of this work seems to be derived from the ElemNet method --- there seems to be a lot of overlap with the authors' paper in *Scientific Reports*. More discussion on what has been gained would help the reader better assess this work.

→ Thank you, we have focused on the deep transfer learning from large DFT dataset (OQMD) to other smaller datasets (JARVIS, Materials Project and experimental observations) in this manuscript which is completely different from our previous work where we demonstrated how we can apply deep learning for automatically capturing chemistry for better prediction modeling using only elemental fractions. We have made it more clear in the introduction section (first paragraph on page 6).

6. Figure 6 (figure 7 in revised draft) and the corresponding discussion are hard to understand – while we can see there is some improvement in the ElemNet-OQMD approach, it does not appear to be highly significant. Some context for how much that difference impacts the selection of chemistries would be helpful to understand the impact of this work.

→ Thank you, the figure is an attempt to explain why transfer learning works. Our goal here is to understand how (rather than 'how much') transfer learning of chemistry learned using OQMD dataset helps in predictive modeling on smaller datasets; if we look at the ROC curves, we can observe high difference in the AUC-ROC values between the models trained with and without using transfer learning, we have added the AUC-ROC values to the legends to make it more clear. The answer to 'how much that difference impacts the selection of chemistries' can be gazed from the improvement in prediction performance after use of transfer learning compared to the model trained from scratch.

7. A small question, but it is discussed that the application of the machine learning model to the smaller experimental datasets provides good results. I was wondering the diversity of chemistries in the experimental dataset – is it more limited than the DFT trained data, thus explaining why it works, or is it able to be applied to as wide of a chemical range.

→ Thank you, since the DFT datasets is large and experimental dataset is small, it is intuitive that the diversity of chemistry in the experimental dataset is more limited than OQMD. However, the experimental dataset used in this work does contain a reasonably diverse set of chemistries which includes oxides, nitrides, hydrides, halides and some intermetallics. Since the DFT datasets is large, it includes all sets of chemistries present in the experimental dataset. As long as the source dataset (OQMD here) contains all the chemical range present in the target dataset (experimental dataset here), we expect the proposed method to work well. We have added this to the dataset and discussion section (page 8, 26).

References:

1. Saal, James E., et al. "Materials design and discovery with high-throughput density functional theory: the open quantum materials database (OQMD)." *Jom* 65.11 (2013): 1501-1509.
2. Kim, George, et al. "Experimental formation enthalpies for intermetallic phases and other inorganic compounds." *Scientific data* 4 (2017): 170162.
3. Kirklin, Scott, et al. "The Open Quantum Materials Database (OQMD): assessing the accuracy of DFT formation energies." *npj Computational Materials* 1 (2015): 15010.
4. Jha, Dipendra, et al. "ElemNet: Deep learning the chemistry of materials from only elemental composition." *Scientific reports* 8.1 (2018): 17593.

Reviewers' comments:

Reviewer #1 (Remarks to the Author):

The authors have made very significant revisions of the manuscript. All of my concerns have been addressed satisfactorily. This is a nice paper and I recommend it for publication in the Nature Communications.

Reviewer #2 (Remarks to the Author):

Thanks authors for addressing my comments in the manuscript. The paper would now be acceptable (a few minor improvements suggested below), except I have a major concern on the validness of the results from the machine learning aspect. This paper cannot be taken into consideration of acceptance until this concern is resolved.

I cannot find the definition of a holdout test set in the paper. The model should be tuned on a validation set for early stopping and the final model should be evaluated on a holdout test set. Even with cross-validation, a holdout test set different from train and validation set is required. However, in both training from scratch and transfer learning tasks presented in the paper, the authors are reporting the error on the validation set as the test error. I am worrying about that the encouraging performance of transfer learning is due to overfitting the validation set.

In order to show the validness of the results, the authors should create holdout test sets for OQMD, JARVIS, Materials Project and experimental data. Whether the model is trained with a simple train/validation split or cross-validation, the model shouldn't be tuned by test set. Finally, the authors should report the performance of the models on the corresponding holdout test sets.

Besides, I have two minor issues:

1. Fig 1. The scatter plots of three datasets have overlap around the 1:1 line. It will be better if the author can split them into three subplots.

2. Line 98 and 109. I prefer authors not to use the term "artificial intelligence" to explain how the network works or why ElemNet perform better than the traditional ML approach. ElemNet (a network of fully-connected layers and dropout) and other traditional machine learning methods are just a subset of "artificial intelligence".

Reviewer #3 (Remarks to the Author):

The authors have addressed my comments. While the work is of value and is well explained, I still have some question as to whether it rises to the impact level that would be expected of Nature Communications. I will however, leave that for the editors to decide.

Response to Reviewer Comments

Manuscript ID: NCOMMS-19-04088

Title: Enhancing Materials Property Prediction by Leveraging Computational and Experimental Data using Deep Transfer Learning

Authors: Dipendra Jha, Kamal Choudhary, Francesca Tavazza, Wei-keng Liao, Alok Choudhary, Carelyn Campbell, and Ankit Agrawal

The authors would like to thank the reviewers for their insightful comments for improving this manuscript. Below we provide point-by-point responses to the comments along with corresponding amendments made in the manuscript.

Reviewer #1 (Remarks to the Author):

The authors have made very significant revisions of the manuscript. All of my concerns have been addressed satisfactorily. This is a nice paper and I recommend it for publication in the Nature Communications.

→ Thank you.

Reviewer #2 (Remarks to the Author):

Thanks authors for addressing my comments in the manuscript. The paper would now be acceptable (a few minor improvements suggested below), except I have a major concern on the validness of the results from the machine learning aspect. This paper cannot be taken into consideration of acceptance until this concern is resolved.

I cannot find the definition of a holdout test set in the paper. The model should be tuned on a validation set for early stopping and the final model should be evaluated on a holdout test set. Even with cross-validation, a holdout test set different from train and validation set is required. However, in both training from scratch and transfer learning tasks presented in the paper, the authors are reporting the error on the validation set as the test error. I am worrying about that the encouraging performance of transfer learning is due to overfitting the validation set.

In order to show the validness of the results, the authors should create holdout test sets for OQMD, JARVIS, Materials Project and experimental data. Whether the model is trained with a simple train/validation split or cross-validation, the model shouldn't be tuned by test set. Finally, the authors should report the performance of the models on the corresponding holdout test sets.

- Thank you for your comment, we have added the result from holdout test sets for all datasets in Table 2 (page 11). We created holdout test sets for all datasets by splitting the original data in two different ratios of 9:1 and 8:2. We trained our models using ten-fold cross validation of the training data and used the best model from our cross validation to report the performance on the holdout test data. The results are in agreement with what we observed from the ten-fold cross validation.

Besides, I have two minor issues:

1. Fig 1. The scatter plots of three datasets have overlap around the 1:1 line. It will be better if the author can split them into three subplots.

-- Thank you, we have split them into three subplots.

2. Line 98 and 109. I prefer authors not to use the term "artificial intelligence" to explain how the network works or why ElemNet perform better than the traditional ML approach. ElemNet (a network of fully-connected layers and dropout) and other traditional machine learning methods are just a subset of "artificial intelligence".

-- Thank you, we have made it more clear by mentioning deep learning.

Reviewer #3 (Remarks to the Author):

The authors have addressed my comments. While the work is of value and is well explained, I still have some question as to whether it rises to the impact level that would be expected of Nature Communications. I will however, leave that for the editors to decide.

→ Thank you.

REVIEWERS' COMMENTS:

Reviewer #2 (Remarks to the Author):

The authors have addressed all issues raised in my reports and the manuscript is now publishable.